# The effect of producer groups on the productivity and technical efficiency of smallholder cocoa farmers in Ghana

**Ebenezer Donkor, Emmanuel Dela Amegbe[☯], Tomas Ratinger[☯], Jiri Hejkrlik[ORCID] ***

Faculty of Tropical AgriSciences, Czech University of Life Sciences Prague, Prague, Czechia

☯ These authors contributed equally to this work.
* hejkrlik@ftz.czu.cz

**Data Availability Statement:** The data underlying the results can be accessed at the following URL: https://data.mendeley.com/datasets/bmwx7tvrsx/1 (DOI: 10.17632/bmwx7tvrsx.1).

## Abstract

Producer groups are influential in Ghana's cocoa value chain. They facilitate training, extension, education and inputs to their members. Still, there is no study on the impact of these producer groups on the technical efficiency and productivity of smallholder cocoa farmers. Using data from 217 and 199 members and non-members of cocoa producer groups, this study aimed to analyse producer groups' impact on smallholder farmers' technical efficiency and yield. The truncated normal distribution stochastic frontier model was adopted to estimate the farmers' technical efficiency. Since the model showed an issue of inefficiency among the farmers, we added socio-demographic and institutional variables to account for determinants of inefficiency. Finally, we adopted an endogenous treatment regression model to analyse producer groups' impact on the farmers' technical efficiency by accounting for observed and unobserved biases. The study results show that farm size, labour, and capital significantly positively impact the partial elasticity of production. Age, education, use of hybrid cocoa, involvement in off-farm jobs, extension access, and producer group membership significantly affect inefficiency. The results further show that producer group membership significantly impacts technical efficiency and yield from the endogenous treatment regression model. To deal with the issues of non-participating in the producer groups, the study recommends that producer groups should be made accessible to farmers. Policymakers can promote the formation and strengthening of producer groups, leading to improved productivity and technical efficiency among cocoa farmers. This approach empowers farmers, enhances their access to resources and knowledge, and enables them to collectively address common challenges, ultimately contributing to sustainable cocoa production and better livelihoods for cocoa farming communities.

## Introduction

Cocoa provides a vital source of foreign exchange for developing countries that produce it and a good source of employment and livelihood for a large portion of their populations [1]. Cocoa production is primarily dominated by poor rural producers and contributes significantly to

**Funding:** The authors wish to acknowledge the Faculty of Tropical AgriSciences and the Internal Grant Agency (20233107) for their financial support of the data collection. The funders had no role in study design, data collection and analysis, decision to publish, or preparation of the manuscript.

**Competing interests:** The authors have declared that no competing interests exist.

the livelihoods of about 40 to 50 million people worldwide [2]. Smallscale farmers (less than 3ha) from Ivory Coast and Ghana are predominant in this business, producing approximately 70% of global production yearly [2]. In Ghana, a third of the population depends on cocoa as their primary source of income [3]. The production and export of cocoa have traditionally dominated the economy, and today Ghana is one of the world's largest exporters of cocoa [4].

Nevertheless, the productivity levels of cocoa remain below maximum capacity, with the average Ghanaian farmer producing only 40 per cent of the farm's potential output [5,6]. On average, Ghana's cocoa yield has been 25% less than the average yield level of the ten largest cocoa-producing nations [7]. Reasons for the low productivity in Ghana include adopting traditional production methods, poor farming practices, and planting low-yielding varieties [8]. Cocoa productivity levels can be enhanced by improving technical efficiency while pursuing sustainable production methods [8].

Enhancing technical efficiency and productivity among smallholder farmers is vital for rural development in many developing countries [9]. Access to information, knowledge, and education is critical [10]. Cocoa farmers need easier access to extension services typically provided by government officials, the private sector, and producer groups. Given that cocoa is cultivated in remote rural areas by many smallholder farmers, the use of producer groups to provide agricultural information, extension, and inputs is more available than government extension agents and the private sector [11].

Producer groups have been regarded by governments, researchers and professionals in international development as possible solutions for gaining sufficient economic size and reducing the transactional costs in rural areas. They can enhance efficient production and marketing and thus overcome the subsistence nature of land cultivation by smallholder farmers, create rural jobs, reduce poverty and improve living standards [11]. Not only the theoretical economic arguments but also many empirical studies highlight a positive relationship between forming and pursuing producer groups' operations and the economic success of smallholder farmers, such as the achievement of higher prices and gross margins and improvement in technical efficiency and productivity [12–16]. However, not much research has been done to ascertain the impact of these producer groups on the productive efficiency and productivity of cocoa farmers. This study, therefore, seeks to bridge the knowledge gap and contribute to the literature.

Establishing producer groups have found its way into the development policy in many developing countries [17]. The government of Ghana has also put numerous strategies in place to support farmers to enhance production through producer groups. The national policy strategies documents such as the Growth and Poverty Reduction Strategy (GPRS II, 2006–2009), the current Medium-term National Development Policy Framework, Ghana Shared, Growth and Development Agenda (GSGDA, 2010–2013), and the Food and Agriculture Sector Development Policy (FASDEP II), highlights the relevance of establishing and strengthening producer groups in developing smallholder agricultural sector in the country. Cocoa farming input and extension services are provided through producer groups that ensure adequate quantities are allocated to farmers and that such allocations are brought promptly to the members. For many donor and NGO programs and projects, producer groups are the only reliable partners in rural areas [11]. Cocoa licensed buying companies (LBCs) have also formed farmer associations to assist producers to enhance productivity through extension services, farm inputs, quality control and other incentives.

For this study, we define producer groups as both cooperatives and farmer associations, as they both provide the same services for cocoa farmers in the Ghanaian context. In general, these groups focus on producing and marketing their products and acquiring farm inputs to improve cocoa output and increase incomes for their members [18]. Other objectives include

an enhanced ability to mobilise savings and attract capital for farm investments, access to labour at low prices for farm maintenance, improved inputs delivery system, and serving as a forum for disseminating cocoa production technologies [18]. The only major difference is that the farmer associations formed by cocoa licensed companies do not fulfil economic cooperative principles since the members have no financial shares in the group, as the group does not exist as a business unit.

Since productive efficiency is the key to achieving cocoa productivity, this study focused on producer group membership's role in ensuring the efficient use of farm inputs to achieve maximum output by smallholder cocoa farmers. In this study, we defined technical efficiency as the ability of a farmer to maximise production with given inputs and a certain technology (output approach) [19]. The rest of the study is organised as follows; section 2 for an empirical review of the study, section 3 for methods and data collection, sections 4 and 5 highlight the results and discussion of the study, respectively, and section 6 provides the conclusion of the study.

## Empirical background

### 2.1 History and a general overview of cocoa production in Ghana

In 1920, Ghana became one of the first cocoa producing countries and by 1930 recorded around 40% of the global production [20]. In 1947, the government of Ghana created Ghana Cocoa Board (Cocobod), a company that was solely in charge of the entire cocoa value chain. It enjoyed monopoly over all cocoa produced in the country and reserved the sole right to buy cocoa beans from producers and sell on the world market. Ghana led in global cocoa production from 1925 to 1976. Between 1970 and 1980, the production dipped when the government decided to increase taxations and expulse thousands of foreigners, who were for the majority employed in cocoa fields: production then fell from 591,000 tons in 1964 to 159,000 in 1983 [21]. Again, in 1983 to 1984 due to aging trees, widespread disease, drought, low producer prices and the alleged smuggling of exportable beans into Côte d'Ivoire, Ghana saw its lowest level of production [21,22].

In the 1990s, Ghana entered a process of semi-liberalization of its cocoa sector. The government introduced robust economic reforms in 1983 and cocoa sector reform in 1993 and 1999 but maintained Cocobod even against recommendations from International Monetary Fund (IMF) and World Bank to dissolve it [23]. The resulting system a "meso-model' of partial liberalization of the cocoa sector which is still in operation today (this model is very close to the pre-liberalization situation in Ivory Coast) [23].

Ghana's partial liberalization has contributed to the revitalization of its cocoa sector and cocoa producers have benefitted from these licensed buying companies (LBCs') competition. Their first way to gain market share was credit services and instant cash payments to producers, which replaced the deferred payments of Cocobod [23]. This eased their penetration into the market that Cocobod had enjoyed monopoly over for so long. With the presence of LBCs, farmers now benefited from cash and carry and credit services instead of future payments by the past monopoly.

Between 2001–2009, Cocobod begun to put in place different programs to sustain farmer production: crop protection program, and a subvention (70%) from 2008 on fertilizers. For some time, cocoa production was relatively profitable, and its production increased [24]. Other forms of response included providing extension services, replacing old cocoa trees with hybrid high-yielding varieties, offering fertilizer subsidies, running mass spraying programs, making improvements to the road network, shifting responsibility for cocoa procurement from the government-controlled Cocobod to privately licensed buying companies (LBCs)

[22]. The Government of Ghana and these privately licensed buying companies embark on extension activities that advocate good farming practices and all other interventions aimed at sustainable production. These stakeholders encourage producer groups for effective and efficient dissemination of information, resources and other support services.

## 2.2 Producer groups in Ghana

Previously, extension theories solely focused on supporting individual farm management and promoting farm-level innovations. However, looking at the challenges of today, many of these exceed the level of individual farms or farm households. Issues like managing collective natural resources, value chain management, collective input supply and marketing, building multifunctional agriculture and venturing into new markets typically require new forms of coordinated action and cooperation among farmers and between farmers and other stakeholders [25]. Producer groups are essential for empowering, alleviating poverty and advancing farmers and the rural poor. Producer groups can also be effective alternatives where the private and public provision of agricultural services have failed. Many innovations involve or depend on the adequate functioning of farmer and community organisations or groups [25].

The renewed interest among both public and private organisations to use producer groups for extension delivery in Ghana is that they enable cost-effective delivery of extension services and empower the producer groups' members to influence policies that affect their livelihoods [26]. Government, donors and partner organisations identify producer groups as having an important role in the country's agricultural development and overall food security [27].

Private sector organisations and LBCs establish producer groups in the cocoa industry to reduce transaction costs and increase profitability. The producer groups enable private entities to deal more effectively and efficiently with smallholder farmers in remote rural areas [28]. Through producer groups, private investors seek to reduce the cost of dealing with farmers, enhance the volume and quality of farm produce, and increase credit recovery rates in farmers' borrowings. They can better provide stable supplies of quality products [29].

The producer groups and their internal dynamics guarantee that various smallholder farmers are educated and trained on good agronomic practices and business practices to enhance safety, quality and productivity [30]. Cocoa farmers in the groups are typically trained on proper agricultural practices, soil management, planting, shade management, pruning, weeding, pest and disease control and prevention, adequate use of fertiliser, proper harvesting techniques and other cocoa cultivation-related activities [31]. Members are trained on safety practices, especially during agrochemical application, proper use of various farm tools and equipment, the dire impacts of child labour and many others [32–34]. This training significantly enhances productivity by informing farmers to make smarter business decisions.

Producer groups are also essential for participating members to attain credit [35] since smallholder farmers have minimal sources of capital for operations and expansions [36]. The risky nature of agriculture and cocoa farming specifically makes financial institutions reluctant to support farmers, and for a good reason. In addition, the lack of appropriate collateral limits smallholder farmers from accessing loans from banks and other financial institutions.

Efficient production in all business forms depends greatly on the kind of inputs entering the production [37,38]. Cocoa productivity relies on the quality of seedlings, the kind of agrochemicals applied, and the tools and equipment available for production, maintenance, harvest and processing [39]. Recently, hybrid cocoa seedlings have been promoted to replace Ghana's old cocoa variety [40]. These seedlings are produced by the Cocoa Research Institute of Ghana (CRIG-CocoBod) and distributed primarily by Seed Production Division—CocoBod (SPD) every year through producer groups. Besides the producer groups, individual farmers can also

request seedlings from SPD. But this is usually at a very high transport cost, so smallholder farmers are discouraged. Every year, various cooperatives and farmer groups receive quality spraying materials in the form of mechanised sprayers, mechanised pruners, appropriate working gear and agrochemicals from government and other stakeholders to undertake mass spraying activities and/or pruning.

However, the effectiveness of extension education through producer groups can depend on various factors, including the quality of the information and advice provided, the level of organisation and management of the groups, and the availability of financial and technical support. Overall, extension education through producer groups can be a powerful tool for promoting sustainable and inclusive agricultural development. By working through producer groups, extension agents can reach more farmers, foster peer learning and exchange, and promote participatory and context-specific approaches to agricultural extension.

## 2.3 Determinants of cocoa productivity and technical efficiency

This section explores the critical determinants of cocoa productivity and technical efficiency, drawing on relevant research studies and empirical evidence.

*1. Farm management practices*: Effective farm management practices significantly impact cocoa productivity and technical efficiency. Adequate planting material selection, proper crop maintenance, and timely adoption of agronomic practices are essential for achieving higher cocoa yields. [41] emphasized the importance of pruning, shade management, and effective weed control in cocoa farms. Furthermore, adopting integrated pest and disease management strategies, including resistant varieties, biocontrol agents, and proper chemical application, is crucial for reducing yield losses [42]. Cocoa trees can become less productive as they age, with yields declining after reaching peak productivity. [43] conducted a study in Nigeria and found that older cocoa trees exhibited lower yields compared to younger trees. They observed a decline in pod production, growth rate, and increased susceptibility to black pod disease as trees aged.

*2. Access to inputs and technology*: The availability and accessibility of quality inputs and appropriate technology are critical for enhancing cocoa productivity and technical efficiency. Access to improved cocoa varieties with high yield potential, disease resistance, and desirable quality traits is crucial. [44] demonstrated that adopting improved cocoa varieties significantly increased productivity and profitability in Ghana. Similarly, access to quality agrochemicals, fertilizers, and irrigation facilities can positively impact cocoa yields [45].

*3. Farmer Education and Training*: Farmers' knowledge and skills play a significant role in cocoa productivity and technical efficiency. Education and training programs targeted at cocoa farmers can enhance their understanding of best practices, sustainable farming methods, and effective management techniques. A study by [46] highlighted the positive impact of farmer training on productivity and income. Training programs focused on integrated soil fertility management techniques. Training programs focused on integrated soil fertility management, pest and disease control, and post-harvest handling can improve cocoa yields and reduce losses [47].

*4. Access to credit and market information*: Access to credit and market information is crucial for cocoa farmers to invest in productivity-enhancing inputs, adopt new technologies, and expand their operations. Studies have shown that limited access to credit is a significant constraint for cocoa farmers, hindering their ability to invest in farm inputs and improve productivity [48]. Moreover, access to market information, price transparency, and linkages to buyers and exporters enable farmers to make informed decisions, negotiate better prices, and reduce post-harvest losses [49].

**5. *Farmer organizations and extension services*:** The presence of well-functioning farmer organizations and effective extension services can contribute to cocoa productivity and technical efficiency. Farmer organizations can provide collective bargaining power, facilitate access to credit and inputs, and enable knowledge sharing among farmers. Studies have shown that farmer organizations are crucial in improving cocoa productivity and profitability [50,51]. Extension services, including agricultural advisory support, training, and dissemination of best practices, are instrumental in enhancing farmers' technical knowledge and skills [52]. Effective extension services can also promote adopting sustainable farming practices and facilitate technology transfer to cocoa farmers.

**6. *Socioeconomic factors*:** Various socioeconomic factors influence cocoa productivity and technical efficiency. Farm size and land tenure systems can affect farmers' incentives and investment decisions. [53] found a positive relationship between farm size and cocoa productivity in Ghana. [54] conducted a study in Ghana and found that farmers with land access and secure land tenure had higher investments in cocoa cultivation, leading to increased productivity and efficiency. Additionally, household characteristics such as education level, gender, and access to social capital can influence cocoa productivity. Studies have shown that educated farmers are more likely to adopt improved practices and technologies, leading to higher cocoa yields [55,56].

Enhancing cocoa productivity and technical efficiency requires a multi-faceted approach considering various determinants. Climate and environmental factors, farm management practices, access to inputs and technology, farmer education and training, credit and market information, farmer organizations and extension services, and socioeconomic factors contribute to cocoa productivity.

## Data and methodology

### 3.1 Study area and data collection

The study area for this study were the Ahafo Ano Southwest and Southeast districts in the Ashanti region of Ghana (Fig 1). The two districts were formerly part of the then-larger Ahafo Ano South district. The districts cover a total surface area of about 1241 km sq., representing 5.8% of the region's total surface area [57]. The districts fall within the forest belt of Ghana. The soil in the districts is generally suitable for agriculture as the vegetation and the climate status are conducive to food production. The soil is deep and can support many cash crops like cocoa, coffee, citrus, oil palm and cola, and food crops like cassava, cocoyam, yam, maize, beans, plantain, rice, sugarcane, and vegetables [57]. It is, therefore, not surprising that agriculture is the highest employer of the population in the district (74.9%) and the third in the Ashanti Region (36.6%) [57].

53% of the employed are engaged in cocoa farming [57]. Almost all the communities in the district are farming communities with an Agricultural Extension Agents (AEAs) ratio of 1:7,604 [57]. With the higher AEA to farmer ratio, producer groups are vibrant in reaching out to cocoa farmers in the area. Training and workshops are organized through Farmer-based organizations (FBOs) or producer groups. Some of the key programs carried out via FBOs in the districts to improve cocoa production are Cocoa Rehabilitation Program, Cocoa Diseases and Pest Control Program (CODAPEC), Cocoa HiTech (Fertilizer) Program, Free Hybrid Cocoa Seedling Distribution, Artificial Hand Pollination, Mass Cocoa Pruning, Cocoa Management System (CMS) [57].

The target group for this study were smallholder cocoa farmers who were members and non-members of producer groups. Multi-stage sampling technique was used for the data collection. Firstly, data was collected by obtaining a list of cocoa producer groups in the Ahafo-

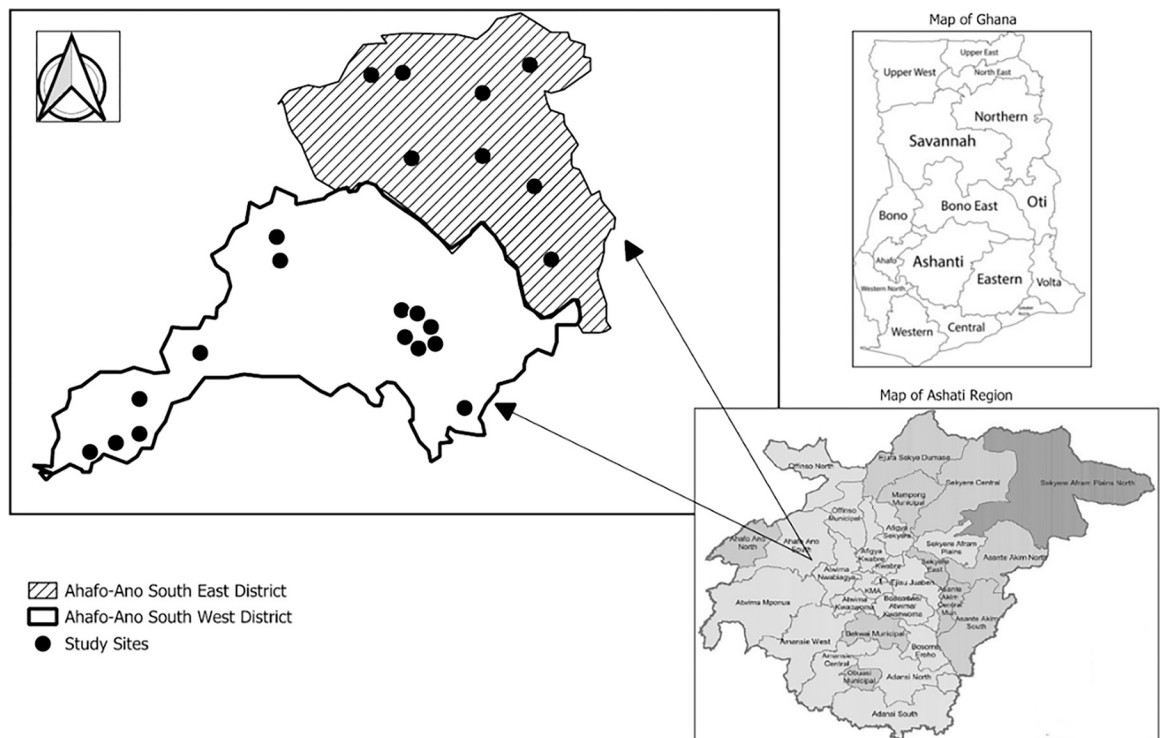

**Fig 1. Map of study area.**

Ano South- East and South-west districts in the Ashanti region of Ghana from the district cocoa cooperative officers. 22 communities were purposively selected from the two districts based on the presence of cooperatives in the communities.

In each producer group, ten members were randomly selected from the list of all members; in total, we interviewed 217. For the non-members, because we didn't have a list of the farmers, we used the purposive and convenience sampling technique to interview farmers in selected communities based on their availability during data collection. The non-members were identified with the help of extension agents and officers of LBCs at the community levels. In total, we interviewed 199 non-members. Therefore, the sample size for the study was 416 farmers.

Data from Cocoa Health and Extension Division (CHED) of CocoBod shows that there is an estimate of 18,688 cocoa farmers in the selected cocoa district. This information was confirmed by the district manager of the Ministry of Food and Agriculture (MOFA). The sample size was thus calculated according to the formula;

$$(z^2*p*(1-p)/e^2)/(1+(z^2*p*(1-p)/e^2*N)) \qquad (1)$$

Where z is the critical value of normal distribution at the required confidence level (1.96 for 5% confidence level), e is margin of error, p is sample proportion which is expressed as decimal, and N is the population size (18,688 cocoa farmers). Considering p as 5% and e as 6%, the sample size would be 377. The sample size for this study was 416 respondents; 219 members and 197 non-members.

Data was collected on farmers' farms, households, institutional characteristics and output of cocoa in the 2020 farming period. To ensure the accuracy and reliability of the cocoa output data, we triangulated the cocoa output data provided by the farmers with the records of the license buying companies officers in the communities since they have records of the quantity

sold by each farmer for the farming period. The data on the traditional inputs for cocoa production (labour, agrochemicals, and farm equipment) were computed in monetary value for data reliability and accuracy purposes. The farmers were able to recall the cost incurred in the farming season on inputs as compared to the quantity they used. Inspiration for collection the traditional inputs in monetary value was obtained from [50]. In addition to the questionnaire data collection, personal interviews were conducted with the producer groups' leaders and Cocobod district officers to understand the support the producer groups provide to the members regarding training, awareness events and inputs. In addition, we interviewed the community officers of the LBCs to understand the productivity issues and support non-members of the producer groups obtained.

## 3.2 Empirical analysis

**3.2.1 Stochastic production frontier.**   This study employed the stochastic frontier model proposed by [58] and extended by [59].

$$\ln(Y_i) = x_i\beta + v_i - u_ix_i = Z_i + \varepsilon_i \tag{2}$$

Here, $Y_i$ is the log value of the output of ith unit; $x_i$ is a vector of exogenous and endogenous variables; $x_i$ is an endogenous variable, $Z_i$ is a vector of all exogenous variables associated with technical inefficiency of the production units, $\beta$ is a vector of unknown coefficient, $v_i$ and $\varepsilon_i$ are two-sided error terms, while $u_i \geq 0$ is a one-sided error term which signposts inefficiency in production. The inefficiency component ($u_i$) is a log difference between maximum attainable output ($y_i^*$) and observed output ($y_i$), namely $u_i = \ln y_i^* - y_i$. The technical efficiency can be expressed as a ratio of observed output ($y_i$) to the frontier output $f(x_i; \beta)exp\{v_i\}$. This is maximum output feasible (with given technology) in an environment with stochastic element $\{v_i\}$. Since $u_i \geq 0$, this ratio lies between 0 and 1. The value 1 signifies that the firm can achieve maximum efficiency. TE$\leq$1 indicates a deficit of output from maximum feasible output within the given condition. This indicates that $vi$ is stochastic and varies across households [60]:

$$TE_i = \exp(-u_i) = \frac{y}{y_i^*} = \frac{y_i}{f(x_i; \beta)exp\{v_i\}} \tag{3}$$

In agricultural production economics research, various mathematical models are used to assess efficiency. Among these models, the most utilized ones are the Cobb-Douglas and translog models. We performed a likelihood ratio (LR) test to compare these two mathematical models. Also, we tested if there is presence of inefficiency in the stochastic production function (Table 1). The LR test statistic was calculated using the formula: LR $= -2\{\ln[H_0] - \ln[H_1]\}$. $H_0$ was determined by the log-likelihood value for the Cobb-Douglas model, while $H_1$ represented the alternative hypothesis and was determined by the log-likelihood value for the translog model. The results of the test indicated that the null hypothesis for the Cobb-Douglas model was rejected at a 5% significance level. This suggests that the Translog Stochastic Frontier Production Function is a more suitable model for analyzing the survey data in the study

**Table 1. Hypothesis testing for functional form selection.**

| Null hypothesis | Test statistic | P-value | Decision |
|---|---|---|---|
| H0: β1 = β2 =. . .. = βn = 0 | Test for Cobb–Douglas vs translog model | | |
| | χ2 (10) = 73.55 | 0.000 | Rejected H0 |
| Sigma_u = 0$\bar{0}$ | Test for absence of technical inefficiency | | |
| | χ2(01) = 77.90 | 0.000 | Rejected H0 |

areas. Consequently, this study utilized the translog function specification to draw conclusions. The likelihood test ratio for the null hypothesis of no technical inefficiency in the stochastic production frontier model is significantly different from zero. The likelihood ratio test also led to the rejection of the null hypothesis H0 at a 5% significance level. This rejection implies that there is indeed inefficiency present in the study areas, and it suggests that the traditional average response function was not a sufficient representation of the data.

The translog function form of stochastic frontier analysis was estimated the model as:Bottom of Form

$$ln(Y_i) = \beta_0 + \sum_{j=1}^{4} \beta_{jk} lnx_{ij} + \sum_{j \leq k=1}^{4} \beta_{jk} lnx_{ij} lnx_{ik} + (v_i - u_i) \tag{4}$$

Here, ln is the natural logarithm, $Y_i$ represents the cocoa output in 2020, $k$ represents the number of inputs used, $x_{ij}$ is input variables (farm size, labour cost, agrochemicals cost and cost of other farm equipment) used by the $ith$ farmer, and $\beta$ is a vector parameter to be estimated. Also, the error term $\varepsilon_i = v_i - u_i$, where $u_i \geq 0$. The random error $v_i$ accounts for stochastic effect is beyond the producers' control, measurement errors, or other statistical noise, and $u_i$ measures the production inefficiency. Based on the previous studies such as [50,54], the variables used to account for inefficiency in this study were the gender of a farmer, age of the farmer, years of formal education, use of hybrid cocoa by the farmer, the average age of cocoa trees, off-farm job involvement, TV/radio ownership as a proxy for a wealthy farmer, access to credit, cocoa land ownership status, and producer group membership.

Cooperative membership decisions are likely to be self-selected, which means it is correlated to inefficiency term $ui$, hence we adopted the endogenous treatment regression model.

**3.2.2 Endogenous treatment regression model.** The endogenous treatment regression model was used to account for selection bias in estimating the impact of producer group membership on technical efficiency. Supposing the impact of producer group membership is $Y_m$, and the endogenous treatment is $t_m$, the outcome equation for the endogenous regression was estimated as follows:

$$Y_m = X_m \beta + \delta t_m + \varepsilon_m, \text{ and } t_m = \begin{cases} 1, if \ w_m \lambda + u_m > 0 \\ 0, if \ w_m \lambda + u_m \leq 0 \end{cases} \tag{5}$$

where $X_m$ are the covariates that affect technical efficiency and yield, and $w_m$ refers to the covariates used to model the producer group membership. The covariates ($X_m$) that were used to model the technical efficiency and productivity were gender, age, TV/radio ownership, years of education, credit access, cocoa land ownership, the average age of cocoa trees, and the use of hybrid cocoa trees, off-farm job involvement, farm size, cost of labour used in the 2020 cocoa farming period, cost of agrochemicals and cost of other cocoa farm equipment in the 2020 farming period.

The covariates ($w_m$) used to model the producer group membership were gender, age, TV/radio ownership, years of education, credit access, farm size, off-farm job involvement, cocoa land ownership, cooperatives in a locality, other group membership and trust. The covariates $x_m$ and $w_m$ are exogenous. $\varepsilon_m$ and $u_m$ are error terms that are bivariate normal with a mean of zero, and the covariate matrix is as follows:

$$\begin{bmatrix} \sigma^2 & \rho^\sigma \\ \rho^\sigma & 1 \end{bmatrix} \tag{6}$$

The likelihood function for observation $m$ of the endogenous treatment regression model was estimated as follows:

$$lnL_m = \begin{cases} ln\phi\left\{\dfrac{w_m\lambda + (y_m - x_m\beta - \delta)\rho/\sigma}{\sqrt{1-\rho^2}}\right\} - \dfrac{1}{2}\left(\dfrac{y_m - x_m\beta - \delta}{\sigma}\right)^2 - \ln(\sqrt{2\pi}\sigma) & t_m = 1 \\ ln\phi\left\{\dfrac{-w_m\lambda - (y_m - x_m\beta - \delta)\rho/\sigma}{\sqrt{1-\rho^2}}\right\} - \dfrac{1}{2}\left(\dfrac{y_m - x_m\beta}{\sigma}\right)^2 - \ln(\sqrt{2\pi}\sigma) & t_m = 0 \end{cases} \tag{7}$$

where $\phi(.)$ is the cumulative distribution function of the standard normal distribution. The ATE estimates from the treatment regression model maximum likelihood estimation can be used as ATT when the outcome is not conditionally independent of the treatment [61]. The instrumental variables used in this study were trust, the presence of the producer group in the farmers' locality and other group memberships such as church and savings group. The study adopted two endogenous treatment models (yield and technical efficiency).

## 4. Results

### 4.1 Description of data

Table 2 highlights the summary of all the variables in this study. The mean difference between the members and non-members as well as T-test are shown in Table 2. The normality test (Shapiro-Wilk and Kolmogorov) showed that the data is normally distributed. The summary statistics results show that the cocoa output in the 2020 farming period is higher for the members than for non-members. Regarding the farmer characteristics, males dominate the members and the non-members. Both the members and the non-members of the producer groups are smallholder farmers, but the farm size of the members is larger than the non-members. The members spend more on agrochemicals than the non-members. The average age of cocoa trees for cooperative farmers is 11 years instead of 14 years for non-members.

### 4.2 Determinants of technical efficiency

From Table 3, the partial elasticities of all the conventional inputs (farm size, cost of labour, cost of agrochemicals and cost of coca farm equipment) influence cocoa output positively. The inefficiency model shows that ownership of TV/radio, years of education, use of hybrid cocoa trees, cocoa land ownership, and producer group membership negatively impact technical inefficiency.

### 4.3 Impact of producer groups' membership on technical efficiency and productivity

Table 4 shows a summary of the yield and technical efficiency of the producer groups' members and non-members. The t-test estimate shows that the members of the producer groups have higher yield and technical efficiency on average than the non-members. Generally, the technical efficiency among Ghanaian cocoa farmers is low. Thus, on average, the members produce only 47.2% of their potential output, whereas the non-members produce only 39.4% of their potential output, given the current technology available to the farmers. The results suggest the prevalence of technical (managerial) inefficiency and little random shocks (climatic changes, production risks etc.) since the estimated gamma is 77.9%. Thus, in the short run, there is a scope for the members and the non-members to increase cocoa production by about 53% and 61%, respectively, if they reduce the technical inefficiencies by adopting new technologies, practices, and efficient combination/allocation of production factors. The results are

**Table 2. Summary of variables used in the study.**

| Variables | Description | Members | | | Non-members | | | Mean Diff. |
|---|---|---|---|---|---|---|---|---|
| | | Min. | Max. | Mean | Min | Max. | Mean | |
| Age | Number of years of respondent | 16.00 | 90.00 | 49.48 (13.82) | 25.00 | 81.00 | 52.62 (46.87) | -3.14 |
| Education | Years of farmer education | 0.00 | 28.00 | 6.31 (5.45) | 0.00 | 32.00 | 6.53 (4.65) | -0.23 |
| Farm size | Size of the cocoa farm (Ha) | 0.26 | 12.50 | 1.95 (1.43) | 0.25 | 6.25 | 1.66 (1.00) | 0.29** |
| Cocoa output | Quantity of cocoa harvested in 2020 season (Kg) | 64.00 | 12800.00 | 1160.15 (77.55) | 64.00 | 3200.00 | 807.70 (37.72) | 352.45 *** |
| Labour cost | The total cost of labour used for cocoa production in 2020 (GHS) | 0.00 | 6000.00 | 718.68 (84.37) | 0.00 | 6000.00 | 1191.57 (105.47) | -472.89*** |
| Cost of agrochemicals | The total cost of agrochemicals used for cocoa production in 2020 (GHS) | 50.00 | 9566.00 | 652.73 (49.92) | 30.00 | 2890.00 | 422.96 (23.78) | 229.77*** |
| Cost of farm equipment | The total cost of cocoa farm equipment used for cocoa production in 2020 (GHS) | 0.00 | 6300.00 | 312.47 (34.68) | 0.00 | 2000.00 | 285.07 (23.61) | 27.40 |
| Age of cocoa trees | The average age of cocoa trees (years) | 1.00 | 17.00 | 11.07 (13.77) | 1.00 | 45.00 | 14.58 (9.04) | -3.52*** |
| Trust | Farmer level of trust for other people in the community (5 points ordinal scale with 5 as highest and 1 as lowest) | 1.00 | 5.00 | 4.21 | 1.00 | 5.00 | 3.92 | 0.29*** |
| | | | Freq. | Percentage | | Freq. | Percentage | |
| Gender | Sex of respondent (male = 1) | | 138 | 63.01% | | 130 | 65.98% | -2.97% |
| Cocoa land ownership | Owner of cocoa farmland (yes = 1) | | 168 | 76.70% | | 106 | 53.80% | 22.90% |
| Ownership of TV/Radio | Farmer-owned radio and television in the house (yes = 1) | | 182 | 83.10% | | 145 | 73.60% | 9.50% |
| Presence of producer group in locality | Presence of producer group in farmer's community (yes = 1) | | 216 | 98.63% | | 127 | 64.46% | 34.17% |
| Off-farm job | Farmer involved in an off-farm business | | 116 | 52.98% | | 85 | 42.85% | 10.12% |
| Hybrid cocoa | Farmer have 30% or more yielding hybrid cocoa trees in the farm (yes = 1) | | 178 | 82.19% | | 169 | 85.78% | -3.59% |
| Credit access | Farmer get access to credit (yes = 1) | | 57 | 26.02% | | 45 | 22.80% | 3.80% |

*** p < .01

** p < .05

* p<. 1, P-values estimated with T-test. The mean values are reported with the standard errors in parentheses. Frequencies and percentages and are reported as indicated.

consistent with other studies from Ghana. [62] study showed 49% technical efficiency on average, and [63] estimated an average of 44% technical efficiency among Ghanaian cocoa farmers.

## 4.4 Endogenous treatment effect estimation

The likelihood ratio test of joint independence is significant at the 1% probability level, showing that the two equations depend on each other (Table 5). The signs and significance of the error of correlation terms indicate that the covariance term of collective producer group membership is statistically significant. Self-selection occurred, hence the relevance of adopting endogenous treatment regression [64,65].

The group membership model shows that farm size, off-farm job involvement, cocoa land ownership status, availability of producer group in farmer's locality and involvement in other groups such as savings groups significantly positively influence producer group membership.

The endogenous treatment regression model indicates that producer group membership significantly impacts farmers' yield and technical efficiency. Apart from producer group membership, ownership of TV/radio, education, use of hybrid cocoa seedlings, and cost of farm equipment positively impacts yield and technical efficiency. The age of the farmer and the cost of agrochemicals positively affect technical efficiency and yield, respectively. However, off-

**Table 3. Maximum Likelihood (ML) estimates of the parameters for the Stochastic Production Frontier (SPF) function and technical inefficiency determinants.**

| Cocoa output | Coef. | St.Err. | t-value |
|---|---|---|---|
| Cocoa farm size | 0.573*** | 0.051 | 11.210 |
| Cocoa farm labour cost | 0.051** | 0.020 | 2.570 |
| Cost of agrochemicals | 0.089** | 0.036 | 2.460 |
| Cost cocoa farm equipment | 0.142** | 0.035 | 4.080 |
| Constant | 1.595*** | 0.327 | 4.880 |
| **Inefficiency** | | | |
| Gender | -0.068 | 0.071 | -0.970 |
| Age | 0.001 | 0.001 | 1.100 |
| Years education | -0.015** | 0.007 | -2.190 |
| Hybrid cocoa | -0.290*** | 0.084 | -3.450 |
| The average age of cocoa trees | 0.002 | 0.003 | 0.860 |
| Credit access | 0.038 | 0.071 | 0.540 |
| Off-farm job involvement | 0.391 | 0.288 | 1.360 |
| TV/Radio ownership | -0.370*** | 0.078 | -4.770 |
| Cocoa land Ownership | -0.108 | 0.077 | -1.410 |
| Producer group membership | -0.148** | 0.064 | -2.330 |
| Constant | 1.261*** | 0.371 | 3.400 |
| /lnsigma2 | -1.254*** | 0.108 | -11.650 |
| /lgtgamma | 0.779 | 0.756 | 1.030 |
| Number of obs | 393 | Chi-square | 248.016 |
| Prob > chi2 | 0 | Akaike crit. (AIC) | 619.18 |

*** p < .01
** p < .05
* p < .1.

farm job involvement significantly negatively influences technical efficiency and yield. The average age of cocoa trees has a negative influence on technical efficiency, while farm size has a negative influence on cocoa yield.

## 5. Discussion

The study investigates the benefits that producer groups offer to their members by establishing the impact of membership on the yield and technical efficiency. To control for other factors, we looked at other characteristics that are important for farmers' yield and technical efficiency.

Our analysis found a positive relationship between farm size and partial elasticity of production but a negative relationship between farm size and cocoa yield. The relationship

**Table 4. Summary of yield and technical efficiency.**

| | Members | Non-members | Mean Diff. | T value |
|---|---|---|---|---|
| Yield (Kg/Ha) | 632.277 (27.804) | 560.030 (23.961) | 72.247** | 1.95 |
| Technical Efficiency | 0.472 (0.012) | 0.394 (0.011) | 0.078*** | 4.75 |

*** p < .01
** p < .05
* p < .1.

**Table 5. Endogenous treatment regression estimates for yield and technical efficiency.**

| | Membership | | | Efficiency | | | Yield | | |
|---|---|---|---|---|---|---|---|---|---|
| | Coef. | St.Err. | t-value | Coef. | St.Err. | t-value | Coef. | St.Err. | t-value |
| Age | 0.001 | 0.003 | 0.220 | 0.001* | 0.001 | -1.890 | -0.517 | 0.495 | -1.040 |
| Gender | 0.081 | 0.153 | 0.530 | 0.026 | 0.016 | 1.560 | 15.109 | 38.231 | 0.400 |
| TV/Radio | -0.221 | 0.216 | -1.020 | 0.099*** | 0.018 | 5.490 | 216.006*** | 42.244 | 5.110 |
| Years of education | 0.01 | 0.014 | 0.710 | 0.004*** | 0.001 | 2.830 | 9.977*** | 3.440 | 2.900 |
| Farm size | 0.13** | 0.062 | 2.100 | -0.009 | 0.007 | -1.410 | -96.115*** | 15.405 | -6.240 |
| Credit access | 0.098 | 0.165 | 0.590 | -0.018 | 0.017 | -1.080 | -55.215 | 38.627 | -1.430 |
| Off-farm job involvement | 0.806*** | 0.370 | 2.180 | -0.182*** | 0.053 | -3.430 | -227.905** | 94.194 | -2.420 |
| Cocoa land ownership | 0.659*** | 0.152 | 4.330 | 0.003 | 0.019 | 0.140 | -34.014 | 44.190 | -0.770 |
| Hybridcocoa | | | | 0.100*** | 0.019 | 5.210 | 171.057*** | 45.771 | 3.740 |
| The average age of cocoa trees | | | | -0.001* | 0.001 | -1.880 | -1.569 | 1.516 | -1.030 |
| Cost of agrochemicals | | | | 0.002 | 0.003 | -0.200 | 0.064** | 0.031 | 2.070 |
| Household labour cost | | | | 0.003 | 0.005 | -0.380 | 0.018 | 0.017 | 1.110 |
| Cost farm equipment | | | | 0.005** | 0.004 | 1.980 | 0.111** | 0.044 | 2.540 |
| Producer group in locality | 2.357*** | 0.298 | 7.910 | | | | | | |
| Other group membership | 0.376* | 0.224 | 1.680 | | | | | | |
| Trust | 0.068 | 0.072 | 0.950 | | | | | | |
| Producer group membership | | | | 0.142*** | 0.033 | 4.330 | 161.614*** | 62.695 | 2.580 |
| Constant | -3.739*** | 0.68 | -5.47 | 0.368*** | 0.065 | 5.650 | 516.276*** | 126.161 | 4.090 |
| athrho | | | | -0.469*** | 0.175 | -2.680 | -0.2* | 0.12 | -1.650 |
| lnsigma | | | | -1.973*** | 0.047 | -41.580 | 5.809*** | 0.04 | 159.30 |
| Number of obs | | | | | | 416 | | | 416 |
| Prob > chi2 | | | | | | 0.000 | | | 0.000 |
| Wald χ2 (10) | | | | | | 148.945 | | | 128.439 |
| LR test of indep. eqns. (rho = 0): | | | | chi2(1) = 6.64 Prob > chi2 = 0.010 | | | chi2(1) = 6.43 Prob > chi2 = 0.011 | | |

\*\*\* p < .01

\*\* p < .05

\* p < .1.

between farm size and partial elasticity of production and yield is complex [66]. The positive relationship between farm size and partial output elasticity can be explained by larger farms' access to more resources (e.g., labour, capital, land), which can increase their ability to respond to input-level changes [67]. For example, a larger farm can hire more workers or invest in new equipment more quickly than a smaller farm, which could increase their productivity in response to input price changes or availability. Additionally, larger farms may benefit from economies of scale, which can increase their production elasticity by reducing per-unit costs [68].

On the other hand, the negative relationship between farm size and productivity is often attributed to diminishing returns to scale [69,70]. As a farm gets larger, it may become more challenging to manage effectively, leading to lower productivity per unit of input. For example, a larger farm may require more managerial oversight, specialised labour, or complex organisational structures, which can increase costs and reduce efficiency.

Labour is a critical input in cocoa production, and increasing labour costs may incentivise farmers to use more efficient labour-saving technologies or techniques to optimise their production processes. Higher labour costs may encourage farmers to increase their investments in human capital, such as training and education, which can lead to higher productivity and

efficiency in cocoa production. By investing in the skills and knowledge of their workers, farmers can improve their ability to manage and optimise their production processes, leading to higher cocoa output. Finally, higher labour costs may incentivise farmers to adopt more sustainable and socially responsible practices in cocoa production. The result is similar [50,71,72].

Agrochemicals and farm equipment are essential inputs in cocoa production. Increasing their cost may incentivise farmers to use these inputs more efficiently or seek out alternative, more cost-effective, alternative inputs. More usage of these inputs leads to a larger increase in cocoa output [50,72].

The stochastic production frontier inefficiency model shows that producer group membership negatively affects cocoa farmers' inefficiency. The endogenous treatment regression models also confirmed that producer group membership positively impacted the efficiency of smallholder cocoa farmers. Studies such as [50,72,73] found a significant positive relationship between producer group membership and the technical efficiency of farmers. The higher technical efficiency among the cooperative members can be linked to the fact that producer group participation offers the cocoa farmers benefits such as access to relevant information regarding cocoa production, access to relevant and government-approved cocoa inputs, and a farmer-to-farmer learning experience among the farmers.

It is clear from the results that, non-group farmers used much labour whereas, group farmers use more capital inputs (more agrochemical and equipment). It is also clear that, group members have better yield and technical efficiency than non-group. However, credit has no statistical impact. It may, therefore, be opined that the members finance their capital cost from either off-farm job income or support from the producer group. Group members have the opportunity to access quality/recommended agrochemicals. In most cases, smallholder cocoa farmers are often isolated and far from agrochemical shops and have minimal chances to buy agrochemicals even when they have money to purchase. But members of various farmer groups can buy agrochemicals and other farm inputs collectively. This reduces transportation costs and, consequently cost per unit. The field visits and the cocoa spraying activities performed by trainers and leaders in the producer groups also increase the chance for the members to be efficient with their cocoa operations. In addition, producer groups can help to improve the quality of the goods they produce. By working together to set quality standards and monitor compliance, they can ensure that their products meet market demand and command higher prices.

The higher age of farmers also contributes to the technical efficiency [19,71]. One reason for this is that older farmers may have accumulated more knowledge and experience over their years of farming, which they can apply to improve their production processes and optimise their use of resources. This knowledge can include practical skills, such as effective crop management techniques and pest control strategies, and a broader understanding of market conditions, weather patterns, and other factors influencing farming outcomes. Moreover, older farmers may have developed better social networks and relationships with other farmers, extension agents, and other support services, which can provide them with access to information, resources, and knowledge-sharing opportunities that can contribute to their technical efficiency [74,75].

The study's results highlighted a positive relationship between farmers' years of education and yield and technical efficiency. Existing evidence also hints at a positive relationship between farmers' years of education and crop yield and technical efficiency [19,76]. Education can provide farmers with the skills and knowledge needed to effectively manage their farms, including how to properly use inputs like fertilisers, pesticides, and herbicides, as well as how to manage irrigation and drainage systems, optimise planting and harvesting practices, and identify and manage pest and disease issues. With higher levels of education, farmers may also

be better able to access information on best practices, new technologies, and market trends, which can help them make more informed decisions and adapt to changing conditions. Education may also enhance farmers' critical thinking and problem-solving skills, which can contribute to their ability to identify and address issues that arise on their farms [77]. The relationship between education and technical efficiency can also be related to access to credit and resources. Farmers with higher levels of education may be more likely to have access to credit and be able to access higher-quality inputs, leading to higher crop yields and improved efficiency [78,79].

Off-farm job involvement affects technical efficiency and productivity negatively. Off-farm jobs can take time and resources away from farm activities, reducing the time and attention farmers can devote to their farms. This can result in reduced productivity and efficiency and decreased attention to essential tasks such as timely planting, irrigation, and pest management. Additionally, off-farm job involvement may lead to farmers neglecting maintenance and investment in their farms as they prioritise their off-farm job. Moreover, off-farm job involvement may reduce farmers' ability to access and apply new technologies and management practices. Farmers who work off the farm may not have the time, energy, or resources to invest in training or to learn about new practices and technologies, which can limit their ability to adopt more efficient and productive farming methods [19,71].

We confirmed that the old cocoa trees cause inefficiency because aged trees cannot produce more cocoa pods [73]. Old cocoa trees typically have lower yields, are more susceptible to pests and diseases, and require more inputs such as fertilisers and pesticides to maintain their productivity. Additionally, older trees can become less responsive to management practices, such as pruning and fertiliser application, reducing yields and lower quality beans. Furthermore, old cocoa trees can also limit the adoption of new technologies and management practices—farmers may hesitate to invest in inputs or new planting materials for old trees with a limited lifespan.

One of the main advantages of hybrid cocoa varieties is their increased resistance to pests and diseases [80]. This can reduce the need for expensive and environmentally harmful chemical inputs, such as pesticides and fungicides, which can increase the profitability of cocoa farming while also promoting more sustainable and eco-friendly farming practices. Also, hybrid cocoa varieties are often more productive than traditional ones, producing more beans per tree and acre.

Tv/radio ownership which served as a proxy for wealthy farmers had a significant positive relationship with yield and productivity. Farmers who own TV/radio may have access to more information on good agricultural practices, market trends, and new technologies, which can help to improve their farming practices and overall productivity [81]. For example, they may have access to agricultural extension services or be more likely to attend training sessions or workshops. Finally, owning a TV/radio may also indicate a higher level of income [82], which can provide farmers with more resources to invest in their farms, such as purchasing inputs or hiring additional labour, which can lead to higher yields and productivity.

## 6. Conclusion

In Ghana, producer groups provide a platform for the sharing of knowledge, cooperation, training and a significant amount of community care. Producer groups in Ghana are frequently regarded as tools for improvements by government, important for international business partners and certification programs. To comprehend the influence of producer groups on smallholder farmers in terms of yield and technical efficiency, this study was conducted. We used a stochastic frontier production model to estimate the technical efficiency of the members

and non-members of the producer groups. The endogenous treatment regression models were used to assess the impact of the producer group membership on technical efficiency by accounting for observed and unobserved bias.

The selection equation model of the endogenous treatment regression model showed a significant positive relationship between producer group membership and farm size, off-farm job involvement, cocoa land ownership status, availability of producer group in farmer's locality, and involvement in other groups.

The results of the stochastic frontier model showed that farm size, cost of labour, agrochemicals and farm equipment have significant positive production elasticities. The stochastic production frontier model also showed that the inefficiency of the cocoa farmers is affected by the wealth of farmers (ownership of TV/radio), years of education, use of hybrid cocoa trees, cocoa land ownership, and producer group membership.

Since education was found to influence inefficiency negatively, education on improving the cocoa farmers' yield should be provided by the extension agents and the producer groups' leaders. Ghana Cocoa Board and the Licensed buying companies can also play a more active role. This could include providing financial support, establishing schools and educational programs in rural areas, and offering scholarships and other incentives for farmers to pursue education.

Finally, the endogenous treatment regression model indicated a significant impact of producer group membership on the efficiency and productivity of the smallholder cocoa farmers. Thus, we confirmed our primary objective of positive relations between membership in cocoa farmers' groups and the efficiency of cocoa production. We encourage the government of Ghana and its development policy of using producer groups to disseminate cocoa innovation and cocoa training to smallholder farmers since participation in producer groups contributes to technical efficiency, and farmers' technical efficiency directly influences smallholder farmers' productivity. However, the situation in Ghana is that many farmers still do not participate in producer groups. To deal with this issue, we recommend making producer groups more accessible with new groups formed closer to the locality of individual farmers. Increasing awareness of the benefits and addressing any perceived opportunity costs or trust issues related to joining is crucial. Tailoring outreach efforts to the specific needs and characteristics of non-members can also help to increase participation. Finally, it is essential to address prevailing cultural and social norms that may discourage collective action and help create a more supportive environment for producer group membership. Ghana Cocoa Board and the Licensed buying companies should provide proper education and awareness to the smallholder farmers concerning the benefits of a producer group membership.

The zone system, which cooperatives in other African countries use to reach more farmers, can be adopted by the producer groups in Ghana as well. The zone system allows the producer groups to train farmers distanced from the main cooperative meeting centre periodically. Each zone has a leader who participates in the producer group meetings and then conveys and trains the farmers according to the information obtained from the meeting.

Also, the farmers who are not owners of the cocoa farm are not likely to join the producer groups. It is evident from the results that farm ownership status has a negative effect on inefficiency. We recommend that the owners of the cocoa farms encourage their caretakers to participate in the producer groups on their behalf, to benefit from the producer groups.

To improve technical efficiency among cocoa farmers in Ghana, we also recommend that the government improve the infrastructure development in rural areas. This could include improving roads and transportation systems to help farmers get their crops to market more efficiently. Policymakers could invest in research and development to develop new technologies and practices specifically tailored to the needs of cocoa farmers. This could include

researching crop varieties more resistant to pests and diseases or more efficient post-harvest handling methods.

## Author Contributions

**Conceptualization:** Ebenezer Donkor, Emmanuel Dela Amegbe, Tomas Ratinger, Jiri Hejkrlik.

**Data curation:** Ebenezer Donkor, Emmanuel Dela Amegbe.

**Formal analysis:** Ebenezer Donkor.

**Methodology:** Ebenezer Donkor.

**Software:** Ebenezer Donkor.

**Supervision:** Tomas Ratinger, Jiri Hejkrlik.

**Validation:** Ebenezer Donkor, Tomas Ratinger, Jiri Hejkrlik.

**Visualization:** Ebenezer Donkor, Emmanuel Dela Amegbe, Tomas Ratinger, Jiri Hejkrlik.

**Writing – original draft:** Ebenezer Donkor, Emmanuel Dela Amegbe.

**Writing – review & editing:** Ebenezer Donkor, Tomas Ratinger, Jiri Hejkrlik.

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
