## [Decision Letter · Decision Letter 0]

7 Aug 2023

PONE-D-23-18150The Effect of producer groups on the Productivity and Technical efficiency of smallholder cocoa farmers in Ghana.PLOS ONE

Dear Dr. Hejkrlik,

Thank you for submitting your manuscript to PLOS ONE. After careful consideration, we feel that it has merit but does not fully meet PLOS ONE’s publication criteria as it currently stands. Therefore, we invite you to submit a revised version of the manuscript that addresses the points raised during the review process.

ACADEMIC EDITOR: The following comments should be taken under consideration in your revised paper:The sequencing of authors in the manuscript is different from the order in the submission system.The corresponding author in the manuscript is not the same as in the submission system.References should be formatted according to PLOS ONE style. You are encouraged to use reference management software such as Mendeley to change referencing style to PLOS ONE’s style.Please use the appropriate statistical formula to determine the required sample size for your study.In table 1 (data summary), please report frequencies beside relative frequencies for categorical variables. For quantitative variables please add minimum and maximum values as well. These information are useful because it give the reader a brief yet a broad information about the collected data.The description of age in table one should be replaced with a better description.Discuss the results of the t-test in table 1 and inform the reader if the conditions of your two sample t-test were met, especially normality and homogeneity of variance.Please mention that you estimated a translog production function and write your model in translag format.Please conduct models selection test to justify the use of translog VS other competing models.

We look forward to receiving your revised manuscript.

Kind regards,

Mohammed Al-Mahish, Ph.D.

Academic Editor

PLOS ONE

Journal Requirements:

2. You indicated that ethical approval was not necessary for your study. We understand that the framework for ethical oversight requirements for studies of this type may differ depending on the setting and we would appreciate some further clarification regarding your research. Could you please provide further details on why your study is exempt from the need for approval and confirmation from your institutional review board or research ethics committee (e.g., in the form of a letter or email correspondence) that ethics review was not necessary for this study? Please include a copy of the correspondence as an ""Other"" file.

Please provide additional details regarding participant consent. In the ethics statement in the Methods and online submission information, please ensure that you have specified (1) whether consent was informed and (2) what type you obtained (for instance, written or verbal, and if verbal, how it was documented and witnessed). If your study included minors, state whether you obtained consent from parents or guardians. If the need for consent was waived by the ethics committee, please include this information.

3. PLOS requires an ORCID iD for the corresponding author in Editorial Manager on papers submitted after December 6th, 2016. Please ensure that you have an ORCID iD and that it is validated in Editorial Manager. To do this, go to ‘Update my Information’ (in the upper left-hand corner of the main menu), and click on the Fetch/Validate link next to the ORCID field. This will take you to the ORCID site and allow you to create a new iD or authenticate a pre-existing iD in Editorial Manager. Please see the following video for instructions on linking an ORCID iD to your Editorial Manager account: https://www.youtube.com/watch?v=_xcclfuvtxQ.

Reviewers' comments:

Reviewer's Responses to Questions

**Comments to the Author**

1. Is the manuscript technically sound, and do the data support the conclusions?

Reviewer #1: Yes

Reviewer #2: Yes

Reviewer #3: Yes

2. Has the statistical analysis been performed appropriately and rigorously? 

Reviewer #1: Yes

Reviewer #2: Yes

Reviewer #3: Yes

3. Have the authors made all data underlying the findings in their manuscript fully available?

Reviewer #1: Yes

Reviewer #2: Yes

Reviewer #3: Yes

4. Is the manuscript presented in an intelligible fashion and written in standard English?

Reviewer #1: Yes

Reviewer #2: Yes

Reviewer #3: Yes

5. Review Comments to the Author

Reviewer #1: The authors studied the effect of producer group soon technoical efficiency of cocoa farmers in Ahoaafo Ano disctricts. The authors used the term 'producer groups'. In cocoa growig areas in Ghana, the producer groups are the same as farmer based on organistions. Tens of studies on tecnhnical effciency of cocoa in Ghana have used farmer-based organisation (producer groups) as an inefficiency variables. This must be appropriately acknowleged. Although you mentiond farmer organisations in page 9 ans supported with Asare et al. (2019), interstingly, oyu never returned to that citation. You many go through some of the studies at this link:https://scholar.google.com/scholar?as_q=technical+efficiency+cocoa&as_epq=&as_oq=&as_eq=Nigeria&as_occt=title&as_sauthors=&as_publication=&as_ylo=&as_yhi=&hl=en&as_sdt=0%2C5&as_vis=1.

To this end, the consideration of producer groups is not a novelty. What is a departure is using farmer-based organisation/producer groups to bifurcate the data or analysis.

On page 20 you discussed the effect of producer groups using studies such as Abebaw & Haile (2013), Onumah et

al. (2013), Ma et al. (2018) and Olagunju et al. (2021). These introduced FBO as an ineffciency variable. Interestingly, whilst Asare et al. (2019) was not used here, only Onumah et al. (2013) is a Ghanaian study in this citations.

I recommend the authors acknowledge that the cocoa producer group is the same as the FBO. Also, the authors must acknowlegde the subtstantial cocoa efficiecny literature that used FBO as an ineffciency variable.

Reviewer #2: This manuscript is very well-written and employs technically sound procedures to describe, analyze and present the data and results. The statistical and econometric modeling is robust and authors have demonstrated good understanding of the theoretical considerations for using these tools in the context of smallholder technical efficiency and productivity studies. The variables span the range of issues associated with smallholder environment in the area of study and the researchers descriptions of the existing constraints dovetails neatly into the urgency to investigate the role of producer groups. The consideration given to selection bias and the approach adopted to address it is consistent with current thinking in the field. The sequencing of the information follows standard research procedures in the social sciences and agricultural economics. I can also add that the subject-matter of the paper is topical in the context of agricultural development in Africa in general and specially in West Africa and Ghana. So, I have a very favourable opinion about the manuscript in terms of the problem context, stated objectives, methodology, and presentation of the results. I recommend it for publication without hesitation.

Reviewer #3: Review Comments to the Author has been uploaded as an attachment.

Overall, this interesting well written study tackles an important issue of how to improve the technical efficiency of coco farmers through encouraging producer group (both cooperatives and farmer associations). I think there is the need for this study to allow research and policy to reach far-reaching conclusions. However, I have several comments that can be considered to enhance the publication potential.

6. PLOS authors have the option to publish the peer review history of their article (what does this mean?). If published, this will include your full peer review and any attached files.

Reviewer #1: No

Reviewer #2: **Yes: **Ajuruchukwu Obi

Reviewer #3: No

---

## [Decision Letter · Decision Letter 1]

7 Nov 2023

The Effect of producer groups on the Productivity and Technical efficiency of smallholder cocoa farmers in Ghana.

PONE-D-23-18150R1

Dear Dr. Hejkrlik,

We’re pleased to inform you that your manuscript has been judged scientifically suitable for publication and will be formally accepted for publication once it meets all outstanding technical requirements.

Kind regards,

Mohammed Al-Mahish, Ph.D.

Academic Editor

PLOS ONE

Additional Editor Comments (optional):

Reviewers' comments:

Reviewer's Responses to Questions

**Comments to the Author**

1. If the authors have adequately addressed your comments raised in a previous round of review and you feel that this manuscript is now acceptable for publication, you may indicate that here to bypass the “Comments to the Author” section, enter your conflict of interest statement in the “Confidential to Editor” section, and submit your "Accept" recommendation.

Reviewer #1: All comments have been addressed

Reviewer #3: (No Response)

2. Is the manuscript technically sound, and do the data support the conclusions?

Reviewer #1: (No Response)

Reviewer #3: Yes

3. Has the statistical analysis been performed appropriately and rigorously? 

Reviewer #1: (No Response)

Reviewer #3: Yes

4. Have the authors made all data underlying the findings in their manuscript fully available?

Reviewer #1: (No Response)

Reviewer #3: Yes

5. Is the manuscript presented in an intelligible fashion and written in standard English?

Reviewer #1: (No Response)

Reviewer #3: Yes

6. Review Comments to the Author

Reviewer #1: (No Response)

Reviewer #3: Pl/s chech the typewriting errors in the references list (for instance reference 3 , 61,... and others).

7. PLOS authors have the option to publish the peer review history of their article (what does this mean?). If published, this will include your full peer review and any attached files.

Reviewer #1: No

Reviewer #3: **Yes: **Mutasim Elrasheed

---

## [Editor Report · Acceptance letter]

16 Nov 2023

PONE-D-23-18150R1 

The Effect of producer groups on the Productivity and Technical efficiency of smallholder cocoa farmers in Ghana. 

Dear Dr. Hejkrlik:

I'm pleased to inform you that your manuscript has been deemed suitable for publication in PLOS ONE. Congratulations! Your manuscript is now with our production department. 

Kind regards, 

on behalf of

Dr. Mohammed Al-Mahish 

Academic Editor

PLOS ONE